# A Numerical Study on the Influence of Strain Rate in Finite-Discrete Element Simulation of the Perforation Behaviour of Woven Composites

**DOI:** 10.3390/polym14204279

**Published:** 2022-10-12

**Authors:** Mohammad Rezasefat, Sandro Campos Amico, Marco Giglio, Andrea Manes

**Affiliations:** 1Politecnico di Milano, Dipartimento di Meccanica, Via La Masa 1, 20156 Milan, Italy; 2PPGE3M, Federal University of Rio Grande do Sul, Porto Alegre 91501-970, Brazil

**Keywords:** perforation behaviour, woven composite, low-velocity impact, strain rate, finite-discrete element simulation

## Abstract

Predicting the perforation limit of composite laminates is an important design aspect and is a complex task due to the multi-mode failure mechanism and complex material constitutive behaviour required. This requires high-fidelity numerical models for a better understanding of the physics of the perforation event. This work presents a numerical study on the perforation behaviour of a satin-weave S2-glass/epoxy composite subjected to low-velocity impact. A novel strain-rate-dependent finite-discrete element model (FDEM) is presented and validated by comparison with experimental data for impacts at several energies higher and lower than their perforation limit. The strain rate sensitivity was included in the model by developing a novel user-defined material model, which had a rate-dependent bilinear traction separation cohesive behaviour, implemented using a VUSDFLD subroutine in Abaqus/Explicit. The capability of the model in predicting the perforation limit of the composite was investigated by developing rate-sensitive and insensitive models. The results showed that taking the strain rate into account leads to more accurate predictions of the perforation limit and damage morphology of the laminate subjected to impacts at different energies. The experimental penetration threshold of 89 J was estimated as 79 J by the strain-rate-sensitive models, which was more accurate compared to 52 J predicted by the strain-rate-insensitive model. Additionally, the coupling between interlaminar and intralaminar failure modes in the models led to a more accurate prediction of the delamination area when considering the rate sensitivity.

## 1. Introduction

Low-velocity impact response of composites has been the focus of many research works in recent years due to their vast use in many engineering fields such as aerospace, automotive, and sporting equipment [1,2,3,4,5]. Composite laminates are susceptible to impacts during their service life, leading to the initiation and propagation of damage [6,7] and reducing their structural performance. Most notably, in the aerospace industries, composites are often subjected to impacts resulting from bird strikes, rocks, or ice collisions [8]. Glass fibre-reinforced polymers (GFRP) are preferred in many engineering applications due to their good mechanical performance and low cost. The complex material constitutive behaviour and multi-mode damage mechanisms of composites still demand more comprehensive studies on their impact performance. Impact perforation has been identified as one of the most deleterious damage-causing phenomena in the study of composite laminates under impact loads since it may trigger most failure modes such as fibre damage (breakage) and pull-out, matrix damage, and delamination, resulting in severe degradation of mechanical properties.

Mohsin et al. [9] performed a numerical and experimental study on the low-velocity impact response of non-crimp fabric-based composites and reported rebounding, penetration, or perforation of the impactor as possible outcomes. Matrix cracking, delamination, fibre matrix debonding, and fibre fracture were identified as probable failure modes at different stages of the laminate response. Perforation occurs when the impact force exceeds the maximum force that the panel can bear, yielding an open force–displacement curve [10]. The penetration and perforation threshold can be obtained using the energy profiling technique [11,12] by performing low-velocity impact tests at different energies to obtain a correlation between impact and absorbed energies. The energy profiling diagram depicts the absorbed energy versus impact energy for all the low-velocity impacts performed in an experimental campaign. This energy profiling diagram can help identify the major damage modes and the energy absorption capacity of the laminates [11]. In a typical diagram, complete perforation of the laminate is identified where absorbed energy is equal to impact energy [11]. For tests with lower energies than the perforation threshold, some of the impact energy is used in the rebound of the impactor. Li et al. [13] investigated the effect of the impactor mass on the low-velocity impact of GFRP and reported catastrophic fibre cracks (longitudinal cracks) as the main failure mode on both impact and rear faces of perforated specimens. Shah et al. [14] used the energy profiling technique to study penetration and perforation of fibre-reinforced composites subjected to single and repeated impacts and found better perforation resistance of a thermoplastic composite, which sustained five impacts at 50 J compared to a thermoset composite, which perforated at the fourth impact.

To study the complex damage mechanism during low-velocity impact, several numerical models have been developed taking into account both interlaminar and intralaminar failure modes of composite laminates [15,16,17,18]. One common approach is to use finite element (FE) models based on continuum damage mechanics (CDM) with progressive failure interlaminar and intralaminar models. CDM-based models can be used for the prediction of the initiation and propagation of failure modes such as fibre breakage, matrix failure, and delamination [16,19,20]. The use of smeared crack approach [21] allows the representation of damage and material degradation while keeping the model computationally efficient [22]; nevertheless, its assumptions require the removal of elements for the simulation of perforation and the models cannot predict stress concentration at the crack tip [23]. The other major drawback of these FE models is the mesh size sensitivity in the prediction of crack orientation [23].

Discrete damage modelling using cohesive zone models (CZM) has been used to simulate connections and separations at predefined interfaces to develop models capable of considering stress concentration around crack tips [24,25,26,27,28]. This way, additional degrees of freedom are added to the model by inserting 2D cohesive elements or spring elements at potential crack paths in between solid 3D elements. The solid elements dominate the global response of the model and the cohesive elements are used to assess the initiation and propagation of damage [24]. Joosten et al. [24] simulated notched composite specimens using a hybrid finite-discrete element model with a combination of solid and cohesive elements to consider discrete cracking. Their results showed a good correlation in damage morphology between the model and experiments. Cao et al. [25] simulated L-shaped cross-ply laminates following a similar approach by using zero-thickness cohesive elements with a bilinear traction separation behaviour between 3D solid elements for the prediction of matrix cracks. Their model was accurate in predicting simultaneous initiation and propagation of damage and kinking cracks observed in the experiments due to the link between failure modes. Bouvet et al. [29] and Hongkarnjanakul et al. [26] presented a discrete 3D model that considered fibre failure in volumetric solid elements and two different sets of zero-thickness cohesive elements for delamination and matrix cracking. Their model showed good capability in assessing the interaction among those failure modes. Similar models were developed by Sun et al. [27] and Soto et al. [28] for the prediction of matrix cracking and delamination by employing cohesive elements. Satisfactory predictions of damage indicated that zero-thickness cohesive elements are suitable to simulate multi-mode failure in a complex load case such as impact. The effect of strain rate on the simulation of composites has been discussed for different composite materials such as polymer [30,31,32] and concrete-based composites [33,34]. May [30] showed that considering the strain-rate dependency significantly alters the impact effect on laminates. This was concluded based on its influence on, namely, interlaminar strength and fracture toughness, which led to the suggestion of including rate dependency in numerical models susceptible to loading rate effects.

In this paper, the perforation of GFRP laminates under low-velocity impact is studied by developing a novel coupled finite element-discrete element method (FDEM). The model contains volumetric solid elements, to account for the impactor/laminate contact and general impact response. In addition, two sets of zero-thickness cohesive elements are used to simulate damage. Delamination or interlaminar failure is considered by defining cohesive elements between composite plies and fibre breakage or intralaminar damage is considered by defining cohesive elements in-between solid elements in each ply. A strain-rate-dependent bilinear traction separation law for the cohesive elements is developed using a VUSDFLD subroutine in Abaqus/Explicit. The results of three models with different fracture energy assumptions, incorporating strain-rate sensitivity, are compared with experimental data, including estimates of the perforation threshold of the laminate.

## 2. Materials and Methods

### 2.1. Experiment Description

This section briefly describes the experimental data set used for verification of the developed numerical model. A detailed description of experimental testing and results can be found in a previous publication [35]. The composite laminate was manufactured with the vacuum infusion technique as described in [35] using satin-weave S2-glass fabrics (302 g/m^2^, 0.24 mm, 22 threads per cm) and AR260/AH260 epoxy resin system. The produced 16-layer laminate had an overall thickness of ≈4 mm and was cut into 150 mm × 100 mm specimens for low-velocity impact tests following ASTM D7136 standard [36]. These tests were performed in a drop-weight impact apparatus in the 18.4 J to 109.7 J energy range achieved by varying the impactor height and mass. The specimens were fixed in a fixture as per the mentioned standard and impacted at the centre. The hemispherical impactor had a diameter of 16 mm and was equipped with a load cell to record the impact response. The impactor was stopped after each test with a pneumatic arm to ensure a single hit in each specimen.

### 2.2. Numerical Model

The FDEM model was implemented by using a combination of 3D solid elements and zero thickness cohesive elements as depicted in Figure 1. An elastic orthotropic behaviour was defined for the 3D solid elements, and the cohesive elements use a bilinear traction separation law. The strain-rate sensitivity included in some models was only considered for the intralaminar cohesive elements. Since the experimental fibre cracks propagated in the warp and weft material directions of the woven fabric, the cohesive elements were accordingly inserted in predefined crack paths. A detailed description of the adopted numerical methodology is presented below.

#### 2.2.1. Strain-Rate Bilinear Cohesive Law

The strain-rate sensitivity for the intralaminar failure modes was considered for material strength and fracture energy. A VUSDFLD subroutine was used in Abaqus/explicit for the implementation of the rate-dependent bi-linear traction separation cohesive law. The bi-linear cohesive behaviour was defined for both normal (mode I) and shear (mode II) failure modes. Equation (1) shows the damage initiation displacement.
(1)$$\delta_{i}^{0}=\frac{\sigma_{i}^{0}}{K_{i}}$$
where *i* = *n*, *s* for normal and shear modes, respectively, $\sigma_{i}^{0}$ is the cohesive strength (maximum traction), and $K_{i}$ is the element stiffness. The final separation ($\delta_{i}^{f}$) is obtained using Equation (2).
(2)$$\delta_{i}^{f}=\frac{2\cdot G_{i}}{\sigma_{i}^{0}}$$
where $G_{i}$ is the fracture toughness for the mixed-mode behaviour. The initiation of damage is governed by a quadratic stress failure criterion according to Equation (3).
(3)$$\frac{\langle \sigma_{n}\rangle {}^{2}}{N^{2}}+\frac{\sigma_{s}^{2}}{S^{2}}+\frac{\sigma_{t}^{2}}{S^{2}}=1$$
where < > is Macaulay bracket that only allows for the damage to grow due to tensile normal traction, and *N* and *S* are the normal and shear cohesive strengths, respectively.

The mixed-mode fracture energy laws of Benzeggagh and Kenane (B-K) [37] govern the propagation of damage. The degradation of the cohesive behaviour is described by the B-K law as:(4)$$G_{n}^{C}+\left(G_{s}^{C}-G_{n}^{C}\right){\left\{\frac{G_{s}}{G_{T}}\right\}}^{\eta }=G^{C}$$
where $G_{S}$ is the out-of-plane direction energy dissipation and $G_{T}$ indicates the total energy dissipation, $G_{i}^{C}$ is the critical fracture energy for normal (*i* = *n*) and shear (*i* = *s*) directions, and η is the material constant.

The flowchart of the VUSDFLD subroutine is shown in Figure 2 in which the vgetvrm function is used to acquire strain components during the explicit solve for each element. The vgetvrm utility subroutine returns the value of the selected variable corresponding to the beginning of the current increment to use in the main VUSDFLD code. Using the necessary strain components at each time increment, the effective strain rate at each time increment is calculated using Equation (5).
(5)$$\dot{\varepsilon }=\frac{\sqrt{{\dot{u}}_{n}^{2}+{\dot{u}}_{s}^{2}+{\dot{u}}_{t}^{2}}}{t_{e}}$$
where $t_{e}$ is the element thickness and $\dot{u_{i}}$ is the velocity corresponding to normal and shear separations. The normal material strength for different strain rates is calculated using Equation (6).
(6)$$\sigma_{n}^{0}\left(\dot{\varepsilon }\right)=\left\{\begin{array}{c} {\left(\sigma^{0}\right)}_{ref\;}\;\;\;\;\;\;\;\;\;\;\;\;\;\;\;for\;\dot{\varepsilon }\leq {\dot{\varepsilon }}_{ref}\\ {\left(\sigma^{0}\right)}_{ref\;}{\left(\dot{\varepsilon }\right)}^{b}\;\;\;\;\;\;\;for\;\dot{\varepsilon }\geq {\dot{\varepsilon }}_{ref} \end{array}\right.$$
where *b* is the material parameters obtained from fitting Equation (6) to strain-rate-dependent input data, and $\sigma^{0}$ is the quasi-static tensile strength. ${\dot{\varepsilon }}_{ref}$ is the reference strain rate which was set equal to 1 s^−1^. Similar strain-rate-dependent material parameters were used in [38,39] for the description of material strength.

#### 2.2.2. Calibration of the Cohesive Law Parameters

The intralaminar strain-rate-dependent material strength was taken from the results of the mesoscale RVE simulation of the same GFRP woven fabric composite by Ma et al. [38]. Figure 3 shows the data and the fitted Equation (6) used to describe the rate-dependent behaviour. The material constant *b* = 0.153 was chosen for the simulations with rate sensitivity. The cohesive element stiffnesses were set to values obtained experimentally for the GFRP based on the ASTM D3039 standard [40].

Two different approaches were used to estimate the rate-dependent fracture energy, as proposed in [40] and illustrated in Figure 4, namely:

(i) Constant fracture energy (CE) approach: The area under the bi-linear curve (shaded in Figure 4a) is defined as the fracture energy and calculated according to Equation (7). The fracture energy is kept constant during the damage evolution for different rate-sensitive yield strengths resulting in lower failure displacements at higher strain rates, as depicted in Figure 4a.
(7)$$G_{c}=\int_{0}^{\delta^{f}}\sigma d\delta =\frac{1}{2}\sigma^{0}\delta^{f}$$

(ii) Constant inelastic failure displacement (CD) approach: The inelastic failure displacement is calculated according to Equation (8). Keeping this displacement constant results in higher fracture energies at higher strain rates, as shown in Figure 4b.
(8)$$\delta_{in}^{f}=\delta^{f}-\delta^{0}$$

It is worth mentioning that the quasi-static mode I fracture energy was set to 30 N/mm [41]. In addition to the two approaches mentioned, another model without the effect of rate sensitivity (NRS) is considered to qualitatively and quantitatively investigate the strain-rate effect in the simulation of low-velocity impact.

#### 2.2.3. FDEM Model Setup

The 3D FDEM model of the GFRP laminae consisted of two parts, as shown in Figure 5. The area in the vicinity of the impactor/laminate contact region, i.e., (25 × 25) mm, was discretised with both 3D solid and zero thickness cohesive elements. Outside that region, just 3D solid elements were used to reduce the computational effort. The FDEM region has a finer mesh compared to the outer region, with an element size of (1 × 1) mm based on the mesh sensitivity analysis reported in Section 3.

As shown in Figure 5, only a quarter of the laminate and impactor were simulated due to the double-symmetry condition. The impactor and the clamping fixture were modelled as rigid parts and the nodes on the corner of the laminate were fixed to prevent out-of-plane deformation mimicking the experimental boundary conditions. The impactor has a mass of 15.58 kg and impacted the laminate with initial velocities that were measured in the experiment. A high friction coefficient of 0.9 between laminate and clamp was implemented by using a penalty-based surface-to-surface contact algorithm in Abaqus/Explicit [42]. The same contact algorithm was used for the interaction between the impactor and the laminate, with a friction coefficient of 0.3 [43].

The solid 3D elements (C3D8R type-8-node, linear brick, and reduced integration element) followed elastic constitutive behaviour with the GFRP properties compiled in Table 1 which were obtained using uniaxial tensile and in-plane shear tests [35,44]. The material properties for the interlaminar cohesive elements are also shown in Table 1, whereas the properties for the intralaminar cohesive elements were discussed in the previous section.

## 3. Results

The implementation of the strain-rate-dependent bilinear traction separation law was verified by performing a simple tensile load case study on a model with a single, zero-thickness, cohesive element inserted between two rigid parts, as shown in Figure 6. The tensile load was applied at different rates including quasi-static (QS) and 125, 250, 500, and 1000 s^−1^. Figure 6a,b compares the results of the constant fracture energy (CE) and constant inelastic failure displacement (CD) models with the mesoscale simulation results in [38]. A good correlation between the rate-dependent normal cohesive strengths with the mesoscale simulation can be observed for both models. The estimated fracture energies at different strain rates are reported in Table 2, and a different trend can be observed, especially at high strain rates. Similar fracture energies at different strain rates were predicted by the CE model, whereas higher values for higher strain rates were predicted by the CD method.

Mesh sensitivity was performed on the tensile model illustrated in Figure 6 to provide a better insight into the applicability of the proposed model. As shown in Figure 7, four different discretisations for the cohesive interface (and related solid mesh) were considered, which consisted of 1, 4, 9, and 16 cohesive elements inserted between the two rigid bodies. The models were subjected to displacement-controlled loading with a rate of 1000 s^−1^. Additionally, in Figure 7, it can be observed that the mesh size did not have a significant influence on the simulation results and the same strength and fracture energy were obtained.

The experimental and numerical force-displacement curves for 109.7 J, 85 J, 72 J, 59 J and 45 J impacts are compared in Figure 8a–d. Perforation was observed for the 109.7 J impact (Figure 8a) and the GFRP laminate was unable to absorb all the impact energy. The failure modes were dominant fibre breakage in warp and weft material directions, delamination, and matrix cracks [14,35]. A sudden drop in the experimental force-time response was observed at the peak force due to the propagation of cracks from the rear of the specimen towards the impact side, producing through-thickness fibre cracks [46]. Figure 8a shows that only rate-sensitive models, i.e., CD and CE, could give an accurate estimate of peak force for the 109.7 J impact and that the model without rate sensitivity (NRS) significantly underestimated the impact peak force. Perforation led to an open force-displacement curve with the impactor having residual velocity after impact. All three models could predict the open force-displacement curves and perforation, and the CE and CD models better correlated the experimental results, whereas the NRS model showed perforation and stiffness loss at an earlier stage.

The results show that the strain rate plays a crucial role in predicting the impact response. Considering the dominance of fibre failure modes for high energy impacts that lead to perforation, the presented numerical framework with strain-rate sensitivity was able to give more accurate predictions of fibre failure. The constant inelastic failure displacement assumption led to higher peak forces and higher impact resistance compared to the constant fracture energy model due to the higher fracture energies at higher strain rates in the former. Additionally important is the more accurate prediction of residual forces after perforation in comparison to CDM models which rely on element removal to simulate perforation [47]. Similar results were observed for lower impact energies in Figure 8, i.e., the NRS model was unable to suitably predict the impact response, whereas the CD and CE models were able to capture the rebound of the impactor for the 45 J and 59 J impacts which correlated well with the experiment. However, for the 85 J and 72 J impacts an open-shaped force–displacement curve was observed, as shown in Figure 8b,c.

The comparison of energy–time curves for 109.7 J, 85 J, 72 J, 59 J, and 45 J impacts are shown in Figure 9a–d, respectively. For impacts leading to penetration or perforation, due to the infinitesimal velocity of the impactor during rebounding/penetration, and the computational limitation regarding the simulation time, the laminate/impactor force did not reach zero in many cases. This can affect the calculation of energy absorbed by the laminate, which is obtained from the force-displacement curve. For the 109.7 J impact, all simulation cases indicate the absorption of impact energy as inelastic energy. The NRS showed a poor prediction of absorbed energy compared to the experimental observations. For perforating impacts at 109.7 J and 85 J, higher energy absorption was observed for the CD model in relation to the CE model due to the higher impact resistance obtained with the CD model. This resulted in a lower residual velocity for the impactor.

This trend was not observed for rebounding impacts at lower energies. For instance, at the 72 J and 45 J energies, a better correlation between the predictions from the CD and CE models was observed, attributed to the smaller significance of the strain-rate effect at lower energy impacts. At 59 J impact, a closed force-displacement curve was predicted by the CD model, while the CE model showed more force drops which required a longer impact time before rebound of the specimen. It is worth mentioning that, in most cases, the NRS model showed perforation of the specimen leading to inelastic energy absorption.

Mesh sensitivity of the low-velocity impact model was studied by discretising laminates with element sizes of 1.5 mm × 1.5 mm (coarse mesh), 1 mm × 1 mm (intermediate mesh), and 0.5 mm × 0.5 mm (fine mesh), as shown in Figure 10a. These models had a total number of cohesive elements equal to 18,800, 29,375, and 169,200 respectively. The comparison between the energy absorption ratio and computational time obtained with the CD model at 45 J impact is presented in Figure 10b. It can be observed that the intermediate mesh size can provide a good balance between accuracy and computational time and therefore this element size was chosen for the simulations.

Table 3 summarises the impact response at different energies and indicates the rebounding and perforating impacts in all studied cases. For the perforated cases at 109.7 J, the calculated error in absorbed to impact energy ratio (E_A_/E_I_) by the NRS model was −39.9%, significantly higher than −7.5% and −1.3% for the CE and CD models, respectively. A greater deviation in peak force was also observed for the NRS model, with a −64.9% error compared to −16.5% and 5.2% for the CE and CD models, respectively. Due to its better predictions, the CD model was selected for the rate-sensitive study. In addition, the capability of the CD and NRS models in predicting damage for the impact at 109.7 J (i.e., at perforation) is further investigated.

The dimensionless energy profiling diagram shows the variations of absorbed energy to impact energy ratio (E_A_/E_I_) with impact energy (Figure 11). A horizontal line was drawn at E_A_ = E_I_, and vertical dashed lines indicate the penetration threshold taken as the average energy between the highest partial perforation case and the lowest full penetration case. Full penetration of the impactor occurred at energies higher than the penetration threshold and rebound of the impactor occurred at lower energies. The experimental penetration threshold was observed at 89 J, which was estimated at 79 J by the CD and CE models and at 52 J by the NRS model, that is, an error of −11.2% and −41.6% for the former and latter models, respectively. This shows that consideration of the strain-rate sensitivity led to an accurate prediction of the perforation threshold by the numerical framework of this work, whereas the NRS model did not show a satisfactory agreement.

Figure 12 shows the cross-sectional contour plots for the 109.7 J impact at different simulation times for the CD model, along with the experimental post-impact cross-section for comparison. The gray lines in this figure represent regions in which the cohesive elements were removed from the simulation due to failure. At t = 1 ms, the first observed failure mode was delamination, as reported in [19] for simulations considering both interlaminar and intralaminar failure criteria in GFRP laminates. The fibre failure initiated in the rear face at around t = 2 ms with complete intralaminar cohesive failure. The fibre cracks propagated towards the impact side and in-plane leading to severe cracks at t = 3 ms. The sudden drop in force–displacement response observed in Figure 8a can be attributed to these fibre cracks and through-the-thickness propagation. Perforation occurs at t = 4–6 ms with continued propagation of interlaminar and intralaminar cracks and dominance of impactor/laminate shear forces. Additionally, in Figure 8a, after t = 6 ms, the impactor moves through the laminate with residual impact velocity and those shear forces cause a slight increase of predicted absorbed energy in Figure 8a.

The progressive failure of the intralaminar and interlaminar cohesive elements of the CD model is shown in Figure 13. Intralaminar damage starts at an earlier stage and leads to the first oscillations and load drops in the force–displacement curve of Figure 8a. Delamination can be observed in all interfaces of the laminate at t = 1 ms. Until t = 3 ms, fast propagation of interlaminar damage is observed which continued by the catastrophic fibre failure at t = 3–4 ms, consistent with the sudden load drop in Figure 8a.

A comparison of experimental and numerical (NRS and CD models) impact and rear face damage morphology at t = 6 ms is shown in Figure 14a,b. In the rear face, a rhombus-like damage pattern can be observed [48,49] in which the longitudinal and perpendicular cracks were aligned with their diagonals. On the impact side, the hole due to the perforation is observed, being similar in shape and dimension for both models to the experimental damage. The longitudinal and perpendicular cracks are also present but with a more local spread which was limited to the contact area. Additionally, matrix cracks are identifiable at the experimental impactor/laminate contact region, but these were not considered in any of the models.

All three numerical models were capable of predicting the longitudinal and perpendicular fibre cracks on the rear side of the specimen. In Figure 14, the NRS model shows longer fibre cracks on the rear side in comparison with the rate-sensitive models. Additionally, the cracks in the impact side spread outside the impactor/laminate contact region. The predictions were more correlated for the CD and CE models with slightly longer cracks for the CD model.

Figure 15 shows crack length vs. impact time for the longitudinal and perpendicular cracks on the rear side of the specimen for the CD and NRS models. Cracks in both directions were initiated sooner with the NRS model. For the CD model, the longitudinal cracks initiated sooner than the perpendicular cracks which were also the same for the CE model. The experimental longitudinal and perpendicular crack lengths were 35.7 mm and 28.5 mm, respectively, representing an error of −10.3% and −8.8% with the CD model, and 17.6% and 33.3% with the NRS model, respectively. It is also observed that adding strain-rate sensitivity to the model led to ≈31% decrease in predicted fibre crack lengths.

Figure 14 also shows the laminate cross-sectional view at t = 6 ms for the NRS and CD models and the experiment. Due to the coupling between intralaminar and intralaminar failure modes, adding strain-rate sensitivity has a significant effect on the prediction of delamination area and shape considering that all models had the same input cards for interlaminar cohesive elements. Figure 14 (cross-sectional images) and Figure 16 qualitatively and quantitatively compare the experimental and numerical delamination observations. Figure 14 shows larger contact gaps between delaminated layers for the NRS model as well as a larger in-plane spread of damage at different laminate interfaces, which is interesting since both models have the same CZM for simulating damage at the interface. This can be attributed to a greater loss of stiffness due to more severe fibre failure in the NRS model, which led to a larger delamination area.

The experimental and numerical results of the total projected delamination area are compared in Figure 16 for the two energies shown in Table 3. The projected area of the failed cohesive elements at the ply interfaces was measured as the numerical delamination area. Elliptical [50] and rhombus-like [48,49] damage patterns have been reported in the literature for woven composites subjected to low-velocity impacts. The rhombus-like damaged area can be appreciated from the experimental results in Figure 16 and is attributed to the shear band failure due to a combination of severe tension and shear stresses and fibre failure on the rear side of the specimen during impact. An elliptical pattern for the total projected delamination area was predicted by the CD and CE models, while the NRS model predicted a rhombus-like pattern which was closer to the experimental observations. As reported in Table 3, a total projected delaminated area of 984 mm^2^ was predicted by the CD model for 109.7 J impact representing a 23.9% error, which was significantly smaller than that for the NRS model (1228 mm^2^, i.,e., 54.7% error). In addition, lower intralaminar fracture energies at high strain rates are observed in the NRS model compared to the CD model resulting in a larger delaminated area. In Figure 16, a more accurate prediction of the damaged area is observed from the CD model compared to that from the NRS model, while the latter provided a better prediction of the damage shape.

## 4. Conclusions

A novel finite element-discrete element method (FDEM) model for the simulation of perforation under the low-velocity impact of GFRP woven fabric laminates was presented. Different intralaminar and intralaminar failure modes and strain-rate sensitivity were considered in the models by developing a user-defined material model in Abaqus/Explicit. The model was validated by single-element simulations and a comparison of force–displacement and energy–time curves with experimental results.

The experimental penetration threshold was equal to 89 J, which was estimated as 79 J with the rate-sensitive models representing a more accurate prediction compared to 52 J from the rate-insensitive model. The laminate was unable to absorb the whole impact energy and perforation was observed for the experimental impact at 109.7 J. All numerical models were capable of predicting perforation and the open force–displacement response curve at this energy. However, without the strain-rate effect, the model was unable to give accurate predictions of peak force (−64.9% error) and absorbed energy (−39.9% error) leading, to a severe underprediction of impact resistance. In contrast, the models which consider strain-rate sensitivity better correlated with the experimental results. In this case, the model with constant fracture energy assumption model (CE) and constant inelastic failure displacement assumption (CD) yielded just −7.5% and −1.3% errors in the prediction of absorbed energy, respectively.

All three models predicted longitudinal (along the specimen’s long side) and perpendicular fibre cracks on the rear side of the specimen and the predictions were more correlated with the experimental observations for the rate-sensitive models. Without rate sensitivity, cracks in both directions were initiated sooner and the fibre crack lengths were significantly longer, and including the rate sensitivity for the intralaminar failure led to a more accurate prediction of the delamination area due to the coupling between intralaminar and interlaminar failure modes. The total projected delaminated area of 984 mm^2^ was predicted by the CD model, with an error of 23.9%, while the rate-insensitive model predicted 1228 mm^2^, with an error of 54.6%, indicating the rate sensitivity also led to satisfactory predictions of the perforation threshold.

## Figures and Tables

**Figure 1 polymers-14-04279-f001:**
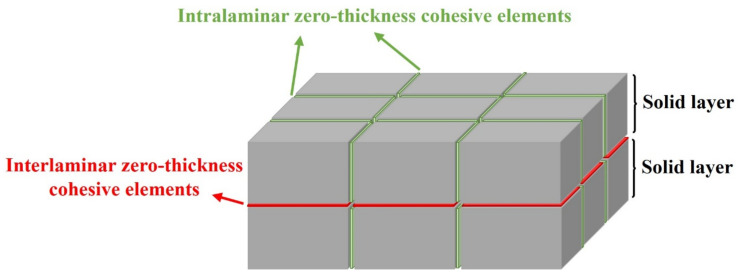
Schematic of the FDEM model.

**Figure 2 polymers-14-04279-f002:**
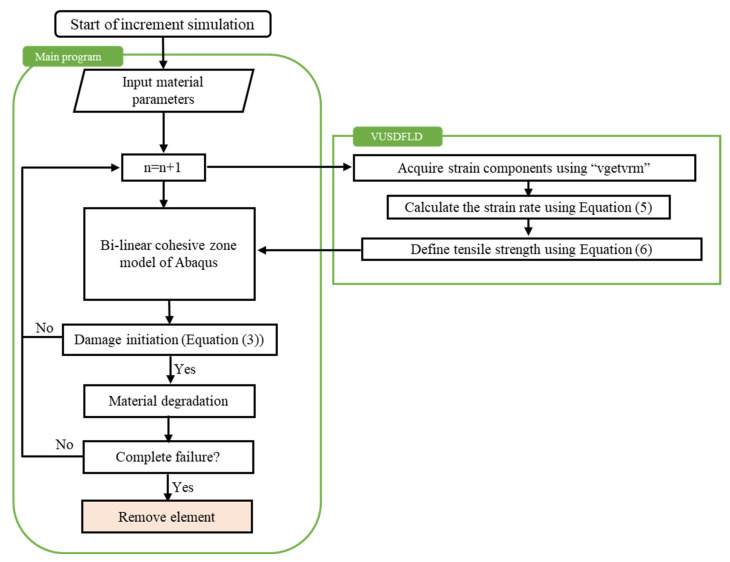
Flowchart of the VUSDFLD subroutine.

**Figure 3 polymers-14-04279-f003:**
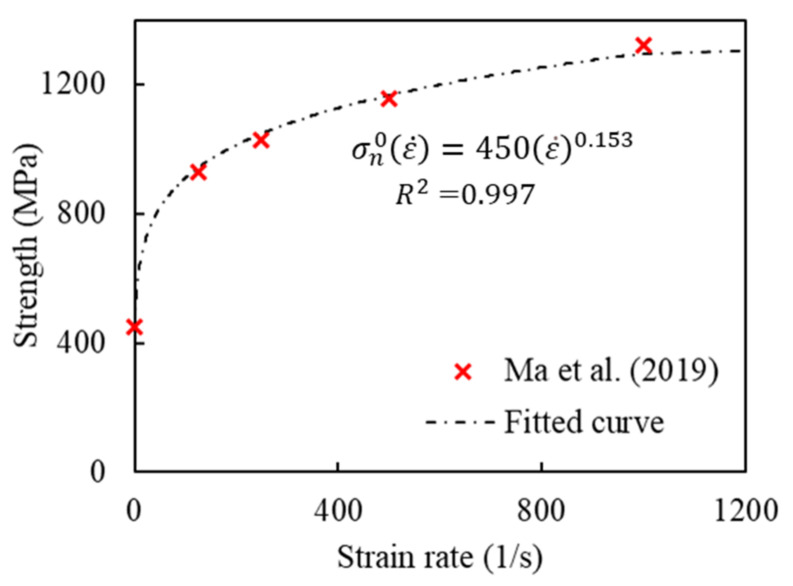
Strain-rate-dependent normal strength estimation for glass fibre woven composites compared with the data from mesoscale RVE simulations of Ma et al. [38]. Drawn on the basis of the data from [38].

**Figure 4 polymers-14-04279-f004:**
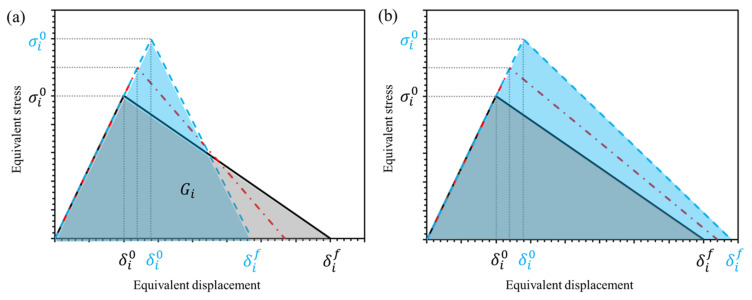
Schematic of bi-linear cohesive behaviour assuming: (**a**) constant fracture energy, (**b**) constant failure displacement.

**Figure 5 polymers-14-04279-f005:**
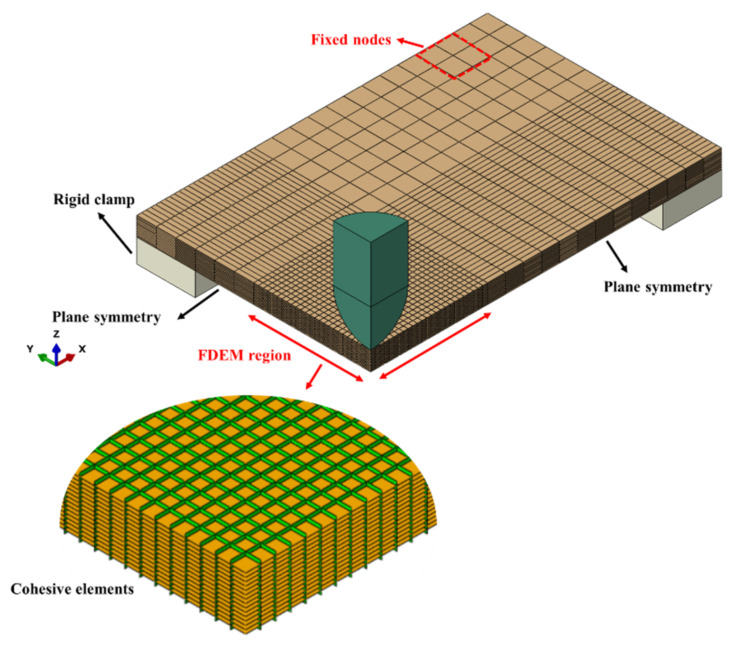
The overall 3D FDEM model with a zoomed view of the cohesive elements.

**Figure 6 polymers-14-04279-f006:**
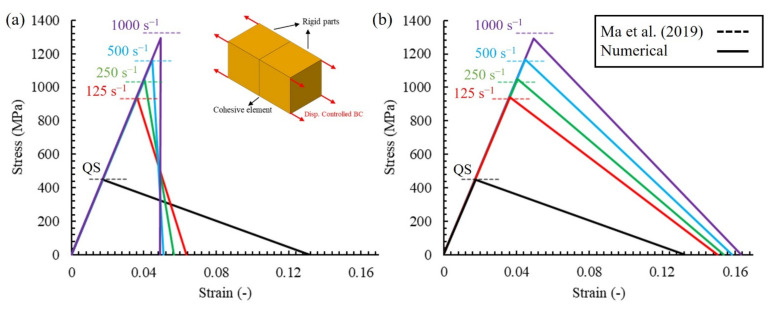
Comparison of the simple tensile loading results at different strain rates with mesoscale simulation results of [38]: (**a**) CE model, (**b**) CD model. Drawn on the basis of the data from [38].

**Figure 7 polymers-14-04279-f007:**
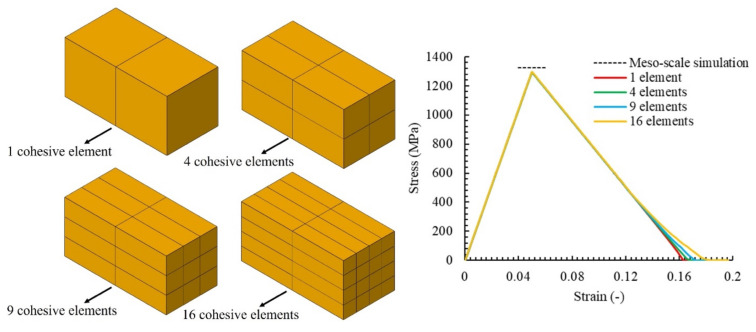
Mesh sensitivity analysis of the tensile load model.

**Figure 8 polymers-14-04279-f008:**
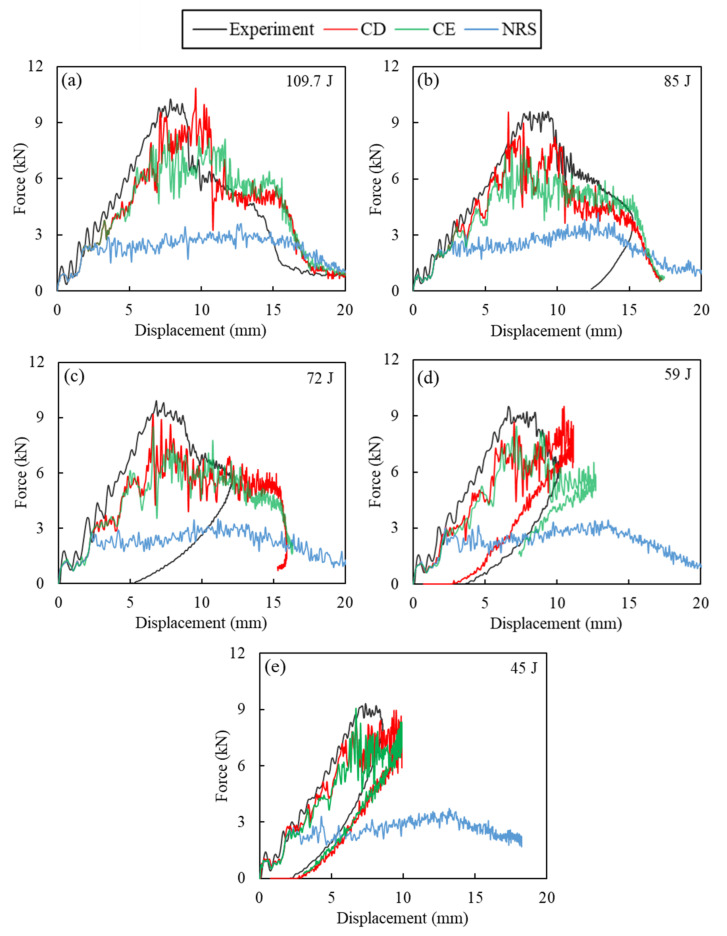
Comparison of numerical and experimental force–displacement curves obtained for impact at: (**a**) 109.7 J, (**b**) 85 J, (**c**) 72 J, (**d**) 59 J, (**e**) 45 J.

**Figure 9 polymers-14-04279-f009:**
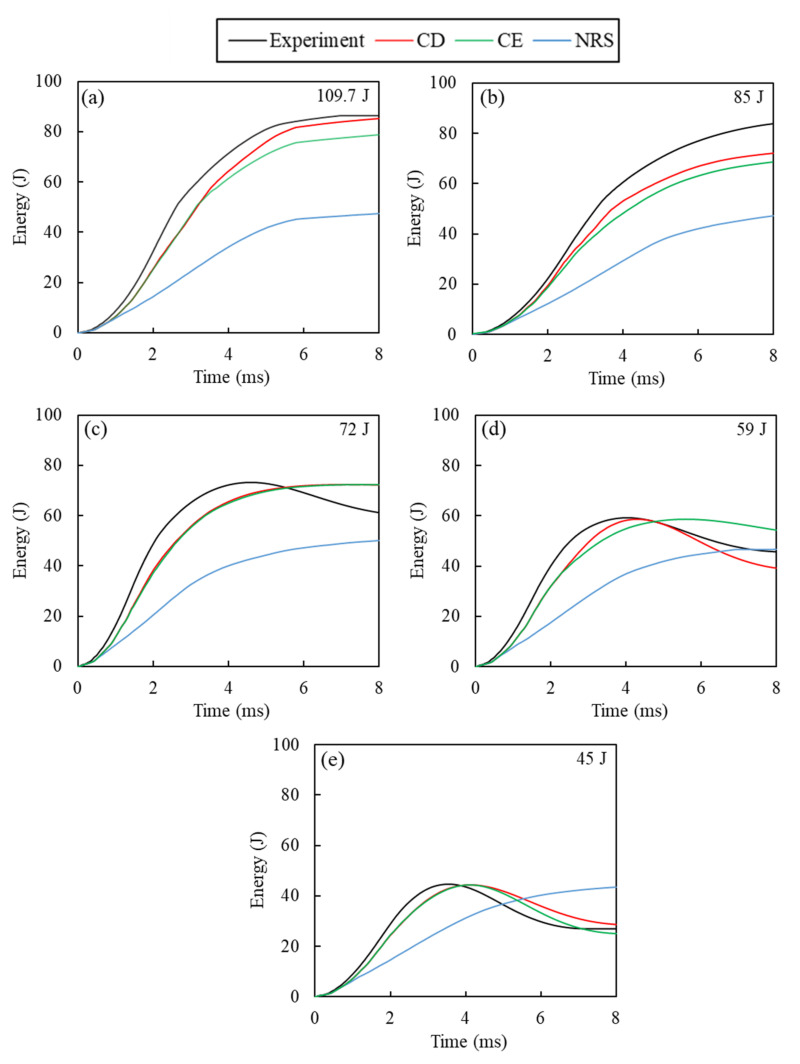
Comparison of numerical and experimental energy-time curves obtained for impact at: (**a**) 109.7 J, (**b**) 85 J, (**c**) 72 J, (**d**) 59 J, (**e**) 45 J.

**Figure 10 polymers-14-04279-f010:**
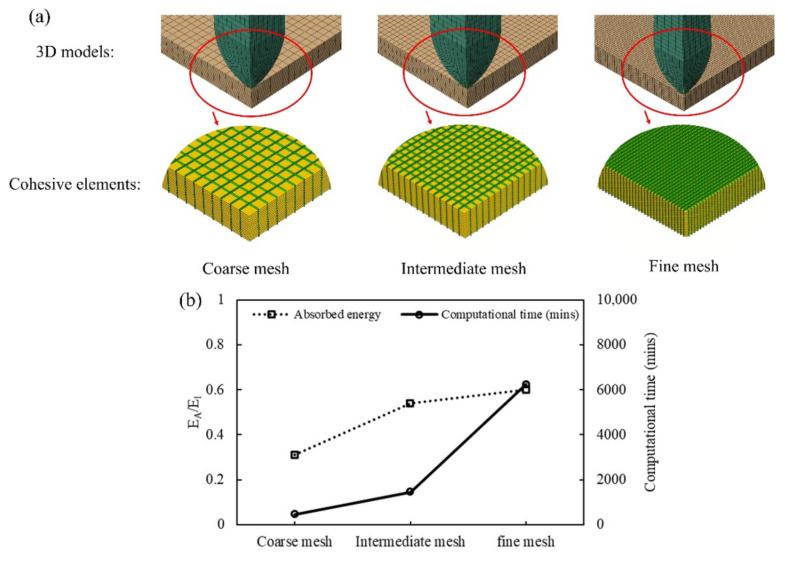
(**a**) Models considered for the mesh sensitivity study of low-velocity impact, (**b**) comparison of energy absorption and computational time for each mesh used.

**Figure 11 polymers-14-04279-f011:**
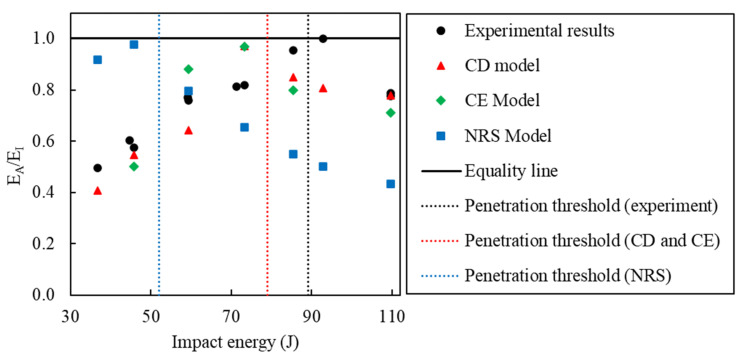
Prediction of perforation threshold from the CD and NRS models in comparison to the experimental results.

**Figure 12 polymers-14-04279-f012:**
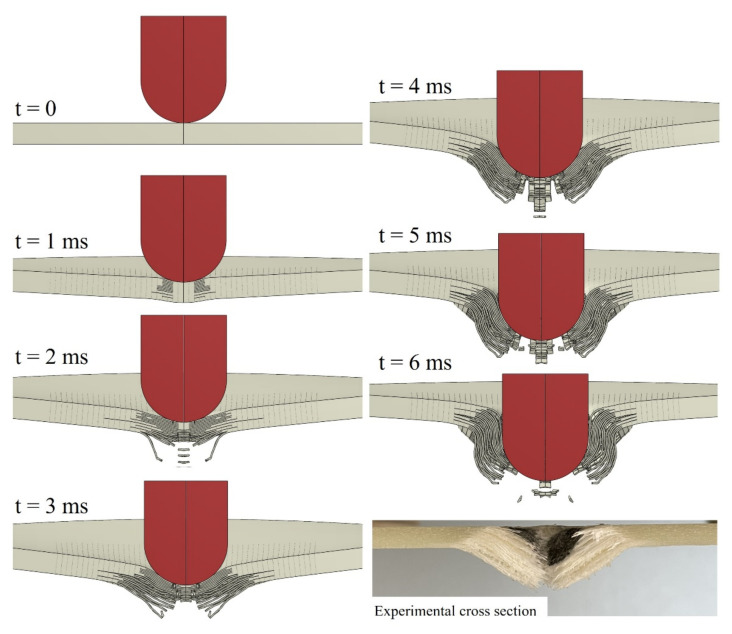
Contour plots of damage morphology at different simulation times for the CD model in comparison to the experimental post-impact cross-section of the specimen.

**Figure 13 polymers-14-04279-f013:**
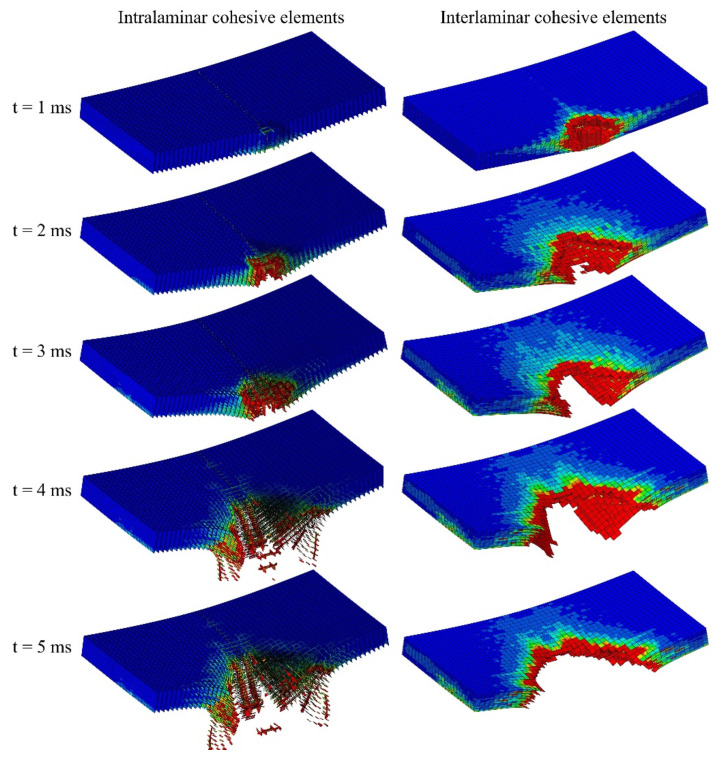
The progressive damage of intralaminar and interlaminar cohesive elements of the CD model. The blue colour indicates no damage, and the red colour indicates a fully damaged region.

**Figure 14 polymers-14-04279-f014:**
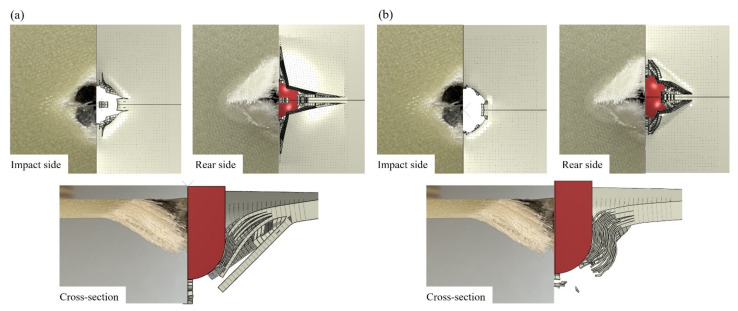
Comparison of the experimental rear side, impact side, and cross-sectional damage morphologies with those from the numerical models: (**a**) NRS and (**b**) CD.

**Figure 15 polymers-14-04279-f015:**
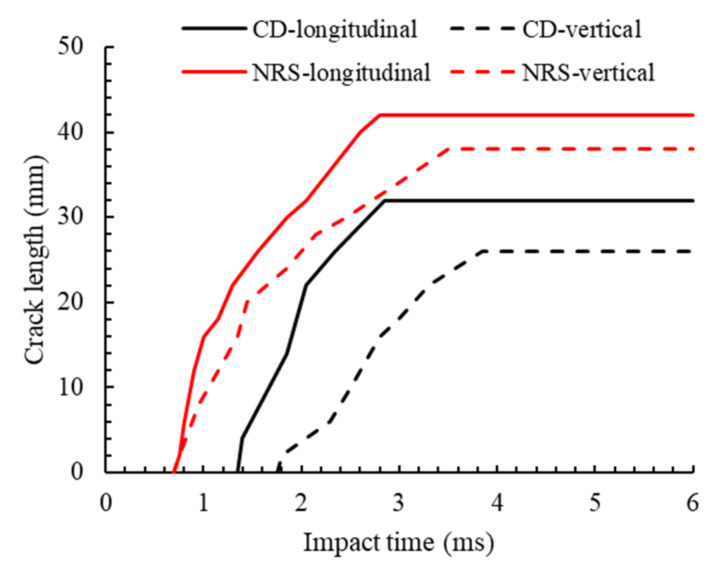
Longitudinal and perpendicular fibre crack lengths versus the impact time for CD and NRS models.

**Figure 16 polymers-14-04279-f016:**
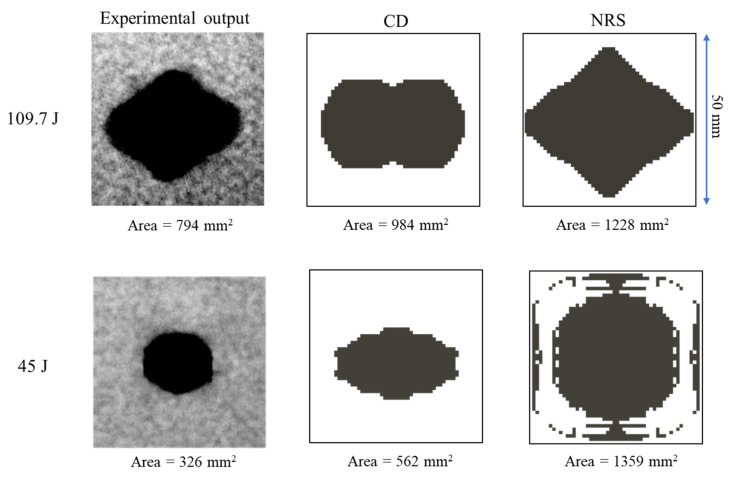
Comparison of experimental and numerical delamination areas for the 109.7 J and 45 J impacts.

**Table 1 polymers-14-04279-t001:** The mechanical properties used for the GFRP plies [35,44] and the interlaminar cohesive elements [45].

GFRP Lamina [35,44]	Symbol [unit]	Value
Density	ρ [kg/m^3^]	1740
Elastic moduli	*E*_11_*, E*_22_ [GPa]	26.5
Elastic modulus	*E*_33_ [GPa]	11.8
Poisson’s ratio	$\upsilon_{21}$	0.12
Poisson’s ratios	$\upsilon_{31}$ $,\;\upsilon_{32}$	0.18
Shear modulus	$G_{12}$ [GPa]	1.80
Shear moduli	$G_{23}$$,\;G_{31}$ [GPa]	2.14
**Interlaminar cohesive** [45]		
Strength	N [MPa]	45.9
	S [MPa]	49.5
Fracture toughness	$G_{n}^{C}$ [N/mm]	0.98
	$G_{s}^{C}$ [N/mm]	3.71

**Table 2 polymers-14-04279-t002:** Fracture energy predictions from the two cohesive zone methods for different loading rates.

Loading Rate (1/s)	Fracture Energy (N/mm)	
	Constant Fracture Energy	Constant Inelastic Failure Displacement
Quasi-static	29.8	29.8
125	29.8	70.8
250	29.7	80.6
500	29.7	92.2
1000	30.8	105.7

**Table 3 polymers-14-04279-t003:** Comparison of experimental and numerical results of peak impact force, absorbed energy, impact energy ratio (E_A_/E_I_), and damage area.

Energy	Response	Experimental Results	Model ID					
			CD	Deviation (%)	CE	Deviation (%)	NRS	Deviation (%)
45 J	Peak Force (kN)	9.61	8.95	−6.7	9.01	−6.2	3.72	−61.2
	E_A_/E_I_	0.57	0.54	−5.2	0.50	−12.3	0.97 **	70.2
	Damage area (mm^2^)	326 *	562	72.4	590	80.9	1359	316.9
59 J	Peak Force (kN)	9.51	9.50	−0.1	8.47	−10.9	3.44	−63.8
	E_A_/E_I_	0.77	0.66	−14.3	0.78	1.3	0.78 **	1.3
	Damage area (mm^2^)	525 *	691	31.6	819	56.0	1486	183.0
72 J	Peak Force (kN)	9.89	9.15	−7.5	8.49	−14.1	3.52	−64.4
	E_A_/E_I_	0.84	0.97	15.5	0.97	15.5	0.65 **	−22.6
	Damage area (mm^2^)	708 *	1104	55.9	1218	72.0	1337	88.8
85 J	Peak Force (kN)	9.60	9.58	−0.2	7.89	−17.8	4.17	−56.5
	E_A_/E_I_	0.95	0.85 **	10.5	0.80 **	−15.8	0.55 **	−42.1
	Damage area (mm^2^)	793 *	1002	26.3	1093	37.8	1303	64.3
109.7 J	Peak Force (kN)	10.2	10.8	5.2	8.5	−16.5	3.6	−64.9
	E_A_/E_I_	0.79 **	0.78 **	−1.3	0.72 **	−7.5	0.43 **	−39.9
	Damage area (mm^2^)	794 *	984	23.9	1036	30.5	1228	54.7

* Experimental damage area was assessed using post-impact backlighting images of the specimens. ** Perforation was observed.

## Data Availability

The data presented in this study are available on request from the corresponding author.

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
