# Peer review of "A Numerical Study on the Influence of Strain Rate in Finite-Discrete Element Simulation of the Perforation Behaviour of Woven Composites"

_polymers, 2022, doi:10.3390/polym14204279_

Round 1

Reviewer 1 Report

This manuscript presents a numerical study on the perforation behaviour of glass/epoxy composite subjected to low-velocity impact by using the strain-rate dependent finite-discrete element model. The strain-rate sensitivity is examined by developing the rate-dependent bilinear traction separation-cohesive model. The perforation limit and damage morphology are accurately predicted for the impacts at different energies. To improve the quality of the manuscript, some comments are suggested as follows:

1.      The language can be improved by correct the sentence structure. In addition, more accurate descriptions and discussions should be made especially for the abstract and conclusion.

2.      There are too many paragraphs in the introduction. The authors are suggested to re-organize the structure by merging some of the related paragraphs.

3.      As a major area of composite materials, the literature review should cover the research on the impact on concrete. One reference is suggested: machine learning method to predict dynamic compressive response of concrete-like material at high strain rates.

4.      The abbreviation is defined for Finite Element-Discrete Element Method (FDEM), this should be used in the later context. So the sentence ‘The coupled finite element-discrete element method of this study was implemented by using a combination of 3D solid elements and zero thickness cohesive elements as depicted in Figure 1’ can be rewritten as ‘The coupled FDEM of this study was implemented by using a combination of 3D solid elements and zero thickness cohesive elements as depicted in Figure 1’.

5.      What is the value of ?̇_??? in the Equation (6) used to describe the rate-dependent behaviour?

6.      Reference errors are found in Section 2.2.3 and Section 3.

7.      Please clarify the mesh sensitivity of the 3D FDEM model with the cohesive elements as shown in Fig. 5.

8.      There are significant discrepancies between numerical and experimental energy-time curves. Please explain and discuss.

Reviewer 2 Report

1.     1. Pg. 1: “Impact perforation … in mechanical properties.” What do the authors mean by the term most important stage? This is not clear here. There can be several different ways in which damage in composite laminates can occur such as delamination, fiber-matrix debonding, fiber fracture, fiber pullout etc. Do the authors mean that impact perforation triggers most of these failure modes?

2.     1. Pg. 2: “Shah et al. [11] … fourth impact.” This sentence is not clear. Please revise.

3.     1. Pg. 3: “The model contains … intralaminar damage.” The reviewer recommends splitting this sentence up for clarity purposes.

4.     1. Pg. 1: Suggestion for the first paragraph: The introduction may be strengthened if the authors can qualify the motivation for studying low-velocity impact response in composite laminates a bit further, i.e., specifying applications and situations wherein impact damage to composite laminates is probable and/or referring to historical scenarios of such impact damage events occurring. However, this is entirely up to the authors.

5.     1. Pg.3: “A strain-rate- dependent bilinear traction separation law …” What is the motivation for using a strain-rate dependent cohesive law? Are the authors speculating that this has a strong effect on the damage behavior? Or is it that during the course of the study, the authors found that strain rate dependency helped in replicating experimental data with the model? If there any studies that have found strain-rate dependent damage events in composite laminate under impact loading, it would be useful to discuss these here.

6.     2.1 Pg. 3: 2nd sentence “The”

7.     2.1 Pg. 3: It may be useful to provide a reference for the standard.

8.     2.1 Pg. 3: What is the justification for using the specific energy range of 18.4 J to 109.7 J?

9.     2.1 Pg. 3: “A detailed description of testing and results can be found in a previous publication [26].” The authors should make it clear that the present work is not describing the experimental setup in its entirety, but only referring to a previous publication in order to compare results in the first sentence of this section. Otherwise, it gives the false impression to the reader that the experiment setup will be described fully in the present paper.

10.  2.2 Pg. 3: What 3D solid elements are being used in the model? 1st order, 2nd order?

11.  2.2 Pg. 3: “The strain-rate sensitivity included in some models was only considered for the intralaminar cohesive elements.” Why is it included only for the intralaminar cohesive elements? Why not the interlaminar cohesive elements?

12.  2.2.1 Pg.3: “It has been shown that considering the strain-rate dependency significantly influences the simulation results of laminates subjected to impact [27].” This sentence would be more appropriately placed in the introduction section. Some discussion on this reference would be useful.

13.  2.2.3. Pg. 7: “A high friction coefficient of 0.9 between laminate and clamp was used [33].” How is this friction coefficient being used? Is this an Abaqus defined contact model?

14.  2.2.3. Pg. 7: Looks like the tables were not referenced properly.

15.  2.2.3. Figure 5: Do all the 3D solid elements in the FDEM region not have connectivity to any of the other solid elements directly? Are there cohesive elements at all interfaces here?

16.  2.2.3. Pg. 7: “The impactor … in the experiment.” The reviewer could not find any discussion about the interaction between the impactor and the laminate. How is this done? Is there a contact model being used here?

17.  3. Pg. 9: “Also, the CD model … in Figures 7b-c.” Couldn’t the same be said of the CE model?

18.  3. Table 3: The results presented here are interesting. If there are experimental results available for the other impact energies, could these also not be presented here, along with their corresponding simulation results?

19.  3. Figure 10: It is not clear what the grey lines represent here. Is it signifying damage? Why does the impacted region turn from a cream color to a grey color from t=0 to t=3ms?

20.  3. Figure 12: Same comment as comment 19.

21.  3. Figure 12: Does the cross-section comparison suggest that the NRS model captures the straight laminate after perforation more accurately? The CD model appears to be showing a more compliant laminate that adheres to the surface of the impactor and curves around it.

22.  3. Pg.15: “fepresenting”

23.  3. Pg. 16: “In Figure 14, a more accurate … from the NRS model.” This sentence is only partially true. Clearly, the NRS model captures the shape of the delamination area much better than the CD model for the 109.7 J impact energy case.

24.  3. Pg. 16: How is the delaminated area being calculated in the model? The reviewer did not find any mention of this in the manuscript.

25.  4. Pg. 17: “bending stiffness.” Was this studied? Can this be quantified?

26.  4. Pg. 17: The last two sentences of the conclusion should be restructured. “Considering” and “Indeed” could be dropped.

Round 2

Reviewer 1 Report

The manuscript has been carefully revised by the authors and can be accepted for publication.